# Knockdown of microRNA-135b in Mammary Carcinoma by Targeted Nanodiamonds: Potentials and Pitfalls of In Vivo Applications

**DOI:** 10.3390/nano9060866

**Published:** 2019-06-07

**Authors:** Romana Křivohlavá, Eva Neuhӧferová, Katrine Q. Jakobsen, Veronika Benson

**Affiliations:** Institute of Microbiology of the CAS, v.v.i., Videnska 1083, 142 20 Prague 4, Czech Republic; romule@volny.cz (R.K.); neuhoferova.eva@gmail.com (E.N.); katrineq@outlook.dk (K.Q.J.)

**Keywords:** nanodiamond, targeted nanoparticles, in vivo application, cancer cell targeting, antimiR, nano-bio interaction

## Abstract

Nanodiamonds (ND) serve as RNA carriers with potential for in vivo application. ND coatings and their administration strategy significantly change their fate, toxicity, and effectivity within a multicellular system. Our goal was to develop multiple ND coating for effective RNA delivery in vivo. Our final complex (NDA135b) consisted of ND, polymer, antisense RNA, and transferrin. We aimed (i) to assess if a tumor-specific coating promotes NDA135b tumor accumulation and effective inhibition of oncogenic microRNA-135b and (ii) to outline off-targets and immune cell interactions. First, we tested NDA135b toxicity and effectivity in tumorospheres co-cultured with immune cells ex vivo. We found NDA135b to target tumor cells, but it binds also to granulocytes. Then, we followed with NDA135b intravenous and intratumoral applications in tumor-bearing animals in vivo. Application of NDA135b in vivo led to the effective knockdown of microRNA-135b in tumor tissue regardless administration. Only intravenous application resulted in NDA135b circulation in peripheral blood and urine and the decreased granularity of splenocytes. Our data show that localized intratumoral application of NDA135b represents a suitable and safe approach for in vivo application of nanodiamond-based constructs. Systemic intravenous application led to an interaction of NDA135b with bio-interface, and needs further examination regarding its safety.

## 1. Introduction

Breast cancer is one of the most frequent women cancers worldwide and results in 13% of total cancer deaths [1]. New strategies for treatment or adjuvant therapies are needed in order to decrease its mortality. In breast cancer as well as other cancers, deregulation of microRNAs enables tumor development and promotes its progression [2,3,4,5]. In order to restore microRNA homeostasis resulting in impairment of tumor growth and its sensitization to conventional therapy, synthetic RNA is introduced into cancer cells using suitable carriers [6,7,8,9]. Such synthetic RNA can be designed to target microRNA overexpressed in tumor cells. This so-called antimiR recognizes sequence of specific microRNA and promotes its degradation.

Nanodiamonds (ND) represent rather promising material for such biomedical application [10,11,12,13]. Their capability to serve as an effective RNA carrier strongly depends on size, shape, preparation method, and on coating of the ND [10,11,13,14,15,16,17]. Especially suitable are the high-pressure and high-temperature (HPHT) ND possessing great biocompatibility [18]. An important feature of these ND is the presence of fluorescent centers (nitrogen-vacancy, N-V) giving them the advantage of traceability [12,14,19].

Even though the HPHT ND have been successfully used in vitro [11,13], there are only a few studies regarding their effectivity in vivo. The existing in vivo studies evaluate organ accumulation and toxicity of naked ND in *Caenorhabditis elegans* [20,21] and mice [22,23]. There is also one study reporting on red blood cells toxicity in human and rats [24]. The in vivo studies agreed the HPHT nanodiamonds were non-toxic and non-harmful even though the nanodiamonds accumulated in lungs, liver, or spleen. The potential accumulation site depends on ND size, coat, and way of administration. While targeting specific tissue such as tumor, we require negligible off-target accumulation. Here, multiple coatings preserving biocompatibility of the ND carrier but driving its cell-specific internalization is needed. Finally, yet importantly, to evaluate the suitability of the ND carriers for topical or even systemic applications, we need more specific applications in vivo showing the different effects of the particular ND-based carrier.

In this work, we focus on HPHT ND with multiple coatings consisting of tumor antigen (transferrin), polymer link (polyethylenimine 800, PEI), and sequence-specific antimiR. All of the four components possess specific function and contribute to the nanocarrier’s (NDA135b) final behavior and effectivity. Our goal was to evaluate if surface coating with tumor specific antigen enables the nanodiamond-based carrier to reach the tumor in a sufficient amount to knockdown particular microRNA and to draft our first idea regarding off-targets and immune cell interactions.

We chose the model of murine breast cancer and we aimed to target oncogenic microRNA-135b that is overexpressed in the cancer cells contributing to the tumor progression and metastasis [25]. First, we successfully applied the NDA135b on cancer cells monocultures in vitro. We assumed then that in vivo, the NDA135b would encounter not only cancer cells, but also immune and endothelial cells. Therefore, we have tested and proved the effectivity of the NDA135b in a mixed co-culture of 3D mammospheres with peritoneal cavity cells, an intermediate model between in vitro and in vivo systems. Importantly, here we found, for the first time, the targeted NDA135b interact with primary granulocytes. We followed with in vivo applications of the NDA135b in a xenograft model of murine cancer and we tested the systemic effects of different ND administrations. The application directly into tumor site as well as systemic intravenous application both led to efficient knockdown of microRNA-135b in tumor tissue. Although, only the intratumoral application resulted in high accumulation of NDA135b in tumors and the NDA135b did not escape into blood. On the other hand, NDA135b applied intravenously were detectable in peripheral circulation and urine. It eventually entered the tumor in a sufficient amount to eliminate microRNA-135b but it also affected the granularity of splenocytes. We believe the intratumoral application of NDA135b represented a more suitable and safer approach for in vivo application of nanodiamond-based constructs.

## 2. Materials and Methods

### 2.1. Materials

#### 2.1.1. Nanodiamond Complex (NDA135b) Preparation

Oxidized fluorescent nanodiamond particles (ND) were obtained from Dr. Petrakova (Faculty of Biomedical Engineering, Czech Technical University, Kladno, Czech Republic) and their preparation has been described in detail by Petrakova et al. [26]. Briefly, the nanodiamond powder (Microdiamant, Lengwil, Switzerland) was purified in a mixture of HNO_3_ and H_2_SO_4_, washed with NaOH, HCl, and water, and freeze-dried. Purified nanodiamonds were irradiated using a 15.5 MeV proton beam, annealed at 900 °C, and air oxidized at 510 °C. Subsequently, the ND were re-purified with HNO_3_ and H_2_SO_4_, dissolved in water (2 mg/mL), and sonicated with a probe (750 W, 30 min). Obtained suspension was filtered via polyvinylidene difluoride (PVDF) membrane with 0.2 µm pores [12]. The obtained ND possessed negative zeta-potential [13]. Before any functionalization, the nanodiamonds were sonicated in the ultrasonic bath for 30 min. Subsequently, the ND solution (1 mg/mL in deionized water) was mixed with equal amounts of PEI 800 (Sigma-Aldrich, Prague, Czech Republic; 0.9 mg/mL) and stirred overnight at room temperature. To prepare a ND complex with targeting structure-transferrin (Tf), we mixed 1 mg of ND (1 mL of water solution) with 200 µg of transferrin conjugate (Alexa Fluor 488 or Texas Red; Life Technologies, Prague, Czech Republic), incubated the mixture in room temperature for 1 h, and subsequently mixed with PEI 800 as described above. Unbound PEI and Tf were removed by centrifugation (9000× *g*, 60 min) and repeated dispersion of the NDs fraction in sterile deionized water in ND concentration 1 mg/mL. (Tf)-PEI-coated ND (1 mg/mL) were kept in a sonication bath, and immediately incubated for 1 h at room temperature with 270 µg of RNA.

A135b, a single-stranded short RNA (antimiR), was designed using the miRBase database (www.mirbase.org) to specifically target and inhibit microRNA-135b. The antimiR sequence was synthesized and modified by IDT (5′ UptCptAptCAUAGGAAUGAAAAGCCptAptUptA 3′, Integrated DNA Technologies; Prague, Czech republic). All the ribonucleotides were modified with 2′ *O*-methyl, and there were six phosphorothioate bonds (pt). Scrambled control RNA (Sc) was used in non-functional complexes (NDSc) instead of A135b. The RNA probes were purified by high-performance liquid chromatography (HPLC). The complexes of ND and RNA were freshly prepared before each experiment and sonicated in a cooled water bath before their use.

#### 2.1.2. NDA135b Complex Characterization

Nanodiamond carriers were characterized by size, surface charge, and load of individual components. We employed dynamic light scattering (DLS) to evaluate size of the complexes. Here, we measured z-average hydrodynamic diameter at a scattering angle of 173° in 25 °C and the overall polydispersity index PdI. Surface charge of the complex estimated as a zeta potential was measured in a clear disposable zeta cells at room temperature. For the measurements and data analyses of zeta potential and DLS, we used Zetasizer Nano ZS (Malvern Instruments, Milcom, Prague, Czech Republic) and Zetasizer Software 7.11 (Malvern Instruments, Milcom, Prague, Czech Republic).

The load of transferrin (Tf) linked to nanodiamond has been determined after the subtraction of unbound Tf from the initial concentration. We used Tf conjugated with Texas Red, and the unbound Tf-Texas Red fluorescence was detected in supernatant using excitation at 595 nm and emission at 615 nm. The concentration of Tf load has been determined from a standard curve of Tf conjugate. The load of PEI was determined by subtraction of unbound PEI from the initial PEI concentration. To detect unbound PEI, we incubated 200 µL of supernatant with 500 µL of 0.150 mg/mL CuSO_4_·5H_2_O (Sigma Aldrich, Prague, Czech Republic) and measured the absorbance at 285 nm. The PEI concentration was calculated from a standard curve of PEI. The load of nucleic acid (NA) was established by subtraction of unbound NA from its initial concentration in reaction. To obtain the unbound NA in supernatant, we pelleted the final complexes 9000× *g* 30 min. NA concentration was measured with a Qubit microRNA assay according to manufacturer protocol (Life Technologies, Prague, Czech Republic). Samples and standards were diluted 1:20 with reaction buffer containing dye (dye dilution 1:1000) and fluorescence was measured using excitation 488 nm and emission 540 nm. Absorbance (PEI quantitation) and fluorescence (Tf and NA quantitation) were detected with an Infinite M200 Pro plate reader (Tecan, Schoeller Instruments, Prague, Czech Republic).

#### 2.1.3. Cell Cultures and Transfection

The 4T1 mammary carcinoma cell line were kindly provided by Dr. Jaroslav Truksa, Institute of Biotechnology, Academy of Sciences, Czech Republic. The 4T1 cells were cultured in RPMI (Sigma Aldrich, Prague, Czech republic) containing 10% (*v*/*v*) fetal bovine serum (FBS, Gibco, Thermo Fisher Scientific, Prague, Czech Republic), 44 mg/L gentamicin (Sandoz, Novartis Company, Prague, Czech Republic), 4.5 g/L glucose, and 1.1% pyruvate (both from Sigma Aldrich) in 37 °C and a humid atmosphere consisting of 5% CO_2_. Cells were passaged three times a week to maintain an exponential growth phase. To obtain 2D cultures, the cells were grown on plastic Petri dishes (TPP, BioTech, Prague, Czech Republic).

Unless stated otherwise, before experimentation, cells were cultured at density 0.5 × 10^5^ cells per well with 1 mL of media (in a 6-well plate, total volume 2 mL) overnight and then stimulated with NDs for 48 h. The final concentration of NDs was always 25 µg/mL corresponding to 6.8 µg/mL of A135b where applicable. The control group (NC) was treated only with PBS.

In the case of 3D spheroid, we cultured 10^3^ 4T1 cells in a 40 µL hanging drop [27] for 10 days and then matured mammospheres were carefully transported into 6-well plate with 2% agarose covered bottoms to underwent stimulations with ND.

Control transfections of A135b without NDs were carried out using X-tremeGENE HP DNA Transfection Reagent (A135bCR, commercial reagent) according to the manufacturer protocol (Roche, Prague, Czech Republic). The ratio of X-tremeGENE HP DNA and A135b was 3:1.

#### 2.1.4. Ex-Vivo Primary Cells Culture and Their Tracking

Cells from peritoneal cavity were obtained by lavage of peritoneal cavity with saline from 12-weeks old Balb/c female mice. The cells were collected by centrifugation and resuspended in RPMI media supplemented with 5% of sera. For cell tracking, total 5 × 10^6^ of peritoneal cells in 3 mL of media were incubated overnight with a 6 µL of CellTracker Violet (Life Technologies, Prague, Czech Republic) in 37 °C and humid atmosphere. In the end of incubation period, the cells were washed with PBS and further used in a co-culture with 4T1 spheroids. When we used a co-culture of peritoneal cells with 4T1 cells, we combined 5 × 10^6^ of peritoneal cells and 2.5 × 10^6^ of 4T1 cells. The co-culture was further maintained (or stimulated) like the 2D cell culture described above.

#### 2.1.5. Quantitative Real-Time-PCR Analysis

The miRNA was purified according to manufacturer instructions using a high pure miRNA isolation kit (Roche, Prague, Czech Republic). After elution, 4 µL of miRNA fraction (total eluted volume 100 µL) were reverse transcribed using specific primers for miR-135b (assay MI0000810) or miR-16 (assay MI0000070) and a high-capacity cDNA Archive Kit (all from Thermo Fisher Scientific, Prague, Czech Republic). PCR quantification was carried out with a TaqMan Universal PCR Master Mix and miR-specific PCR primers (Thermo Fisher Scientific, Prague, Czech Republic). MiR-16 was used as an internal control for miR quantitation. During qPCR, all samples were analyzed in triplicate using an iQ5 Real Time PCR Detection System (BioRad, Prague, Czech Republic). The obtained data were analyzed using iQ5 Optical System Software 2.1 (BioRad, Prague, Czech Republic). The expression of the miR-135b was normalized to the expression of miR-16 and fold change was calculated by the software (based on 2^−ddct^ method).

#### 2.1.6. Lactate dehydrogenase (LDH) Assay

Lactate dehydrogenase released from damaged cells was used to evaluate direct cytotoxicity. The cells were seeded in 96-well plates (5 × 10^3^ cells per well) in triplicates and stimulated the next day with ND complexes for 48 h. Culture media was supplemented with 1% FBS as suggested by manufacturer. A commercial lysis buffer was added to one triplicate of the control cells for the last 3 h of the experimental period (serving as an LDH assay positive control, PC). Negative control comprised from non-stimulated cells (NC).

The LDH assay was performed according to manufacturer instructions (LDH assay, Roche, Prague, Czech Republic). Briefly, when the experiment completed, a cell supernatant was collected and incubated with LDH dye for 10 min. Then, the absorbance was measured with an Infinite M200 Pro plate reader (Tecan, Schoeller Instruments, Prague, Czech Republic) using specific excitation at 490 nm and a reference excitation at 630 nm.

#### 2.1.7. Flow Cytometry

The apoptosis of 4T1 cells was assessed using an Annexin V Dyomics/Hoechst33258 staining (Exbio Antibodies, Vestec, Czech Republic) and the assay was performed according to manufacturer’s protocol. Cells positive for Annexin V and Hoechst33258 were detected by a BD LSR II flow cytometer (BD Bioscience, Prague, Czech Republic). The settings were following: Annexin V 473/520 nm and Hoechst 405/461 nm.

The surface expression of transferrin receptor has been assessed with a transferrin (Tf) conjugated with Texas Red. One million cultured cells were incubated with 1 µL of transferrin-conjugate (1 µg/mL) for 30 min and co-stained with Hoechst 33258 for 15 min. Settings for Tf detection: Ex/Em 559/615 nm. The same settings were used to detect Texas Red in cells obtained from homogenized tissues in the end of in vivo experiments.

The percentage of positive cells for each fluorophore was analyzed by a FlowJo v9.9.4 software (BD Bioscience, Prague, Czech Republic).

#### 2.1.8. Confocal Imaging

For confocal analysis, the 4T1 cells were plated onto 6-well glass bottom plates (10^5^ cells/well) and incubated with functionalized ND for the assay-specific time points. At the end of the incubation period, cell nuclei were stained with Hoechst33342 (1 µg/mL; Invitrogen, Life Technologies, Prague, Czech Republic). In the case of spheroids, the mammospheres were transferred to a glass-bottom dish. The images were acquired with an Olympus FluoView FV1000 confocal microscope (Olympus, Prague, Czech Republic; objective 20×/NA 0.75; 40×/NA 0.95; and 60×/NA 1.35). The data were analyzed with Olympus FluoView 2.0 software (Olympus, Prague, Czech Republic). Excitation/emission parameters were as follows: Hoechst 405/461 nm; NDs 559/655–755 nm; Alexa Fluor 488 473/520 nm; Cell Tracker Violet 415/516 nm. Photo-bleaching (with a 405 nm laser) was used to distinguish a non-specific and the ND-specific signal. In order to imagine the sample depths (particularly spheroids) we performed z-stack imaging with steps of 5 µm.

#### 2.1.9. Animal Model and In Vivo NDA135b Application

Animals’ procedures followed the Czech law regarding animal protection and the Czech Academy of Sciences with number 82/2015 approved the experimental plan. To develop breast tumor, we inoculated 10^6^ 4T1 cells (in saline) into fads pads of six-week-old female Balb/c mice (day 1). Mice were kept under standardized conditions and the tumor development was observed carefully. We started treatment with NDA135b on day 14 when a palpable tumor with a diameter of about 0.5 cm developed. The animals were divided into several groups (3 animals per group): One control group, two groups with NDA135b administered via tail vein, and two groups with NDA135b administered directly into tumor tissue. The control group received saline intravenously. Animals administered with NDA135b received NDA135b with transferrin conjugated (i) with Alexa Fluor 488 or ii) Texas Red. The animals obtained 20 µg of ND carrying 5.4 µg of A135b per mouse for three consecutive days. One hour after the third dose, we took peripheral blood and urine from anesthetized animals, the animals were sacrificed, and different tissues were collected (liver, spleen, kidney, and tumor) for subsequent analyses of NDA135b presence (enabled by transferrin-fluorophore conjugates) and antimiR-135b effect. The in vivo experiments were repeated twice.

#### 2.1.10. Spectrometer Measurements of NDA135b in Fluids

Blood or urine from experimental animals were diluted five times with saline and 100 µL of fluid was placed in the 96-Well optical-bottom plates (Nunc, Thermo Fisher Scientific, Brno, Czech Republic). Fluorescence of Texas Red was measured using 595/615 nm filters and Infinite M200 Pro plate reader. The amount of complexes in the experimental sample was calculated from standard curve of NDA135b samples with known concentration.

#### 2.1.11. Whole Tissue Imaging

At the end of experimental period, the excised liver, spleen, lung, and tumor tissues were imaged with a fluorescent microscopy OV-100. The microscope setup enables to read transferrin-Alexa Fluor 488 (Tf-A; settings 473/520 nm). After visualization, the organs were homogenized, cells were separated via 70 µm filters, and used to detect transferrin–Texas Red by flow cytometry (settings described earlier).

#### 2.1.12. Statistical Analysis

If not stated otherwise, the results are presented as means ± SD of three independent experiments. Statistically significant differences in the tested parameters were assessed using a two-tailed *t*-test with a confidence interval of 95% for paired comparison or ANOVA/post-hoc Tukey for group comparison (online tool available at http://astatsa.com/OneWay_Anova_with_TukeyHSD/). Values of *p* ≤ 0.05 (*) and *p* ≤ 0.01 (**) were considered to be statistically significant between the compared groups.

## 3. Results

In order to properly evaluate the application possibilities of NDA135b complex, we have employed different experimental systems starting with the simplest monoculture of adherent cells derived from mouse mammary tumor. Subsequently, we performed tests in advanced model such as mixed 3D co-culture imitating tumor microenvironment. Finally, we tested NDA135b in vivo, employing local as well systemic administration. Schematic composition of NDA135b and models used in this study are depicted in Figure 1.

### 3.1. Characterization of the NDA135b Complex

In neutral pH, the complete NDA135b construct possessed zeta potential of −20 ± −3.42 mV, size of 353 ± 140 nm, and polydispersity index of PdI = 0.4. The PdI measured by DLS for a uniform sample equals zero and the value 0.4 represents moderate polydispersity due to transferrin–fluorophore conjugate. In comparison to the complex NDA135b, we tested plain ND and ND coated with NA but lacking Tf. The comparison is shown in Table 1.

During the NDA135b preparation, we measured the amount of Transferrin, PEI, and A135b load. One milliliter of the NDA135b complex consisted of 1 mg of ND, 191 µg of transferrin conjugate, 740 µg of PEI800, and 270 µg of antimiR-135b RNA.

### 3.2. Effective Delivery of NDA135b into Adherent Breast Cancer Cells

Using conventional 2D culture of 4T1 mammary tumor cells enabled us to characterize the effect of antisense RNA (A135b) coated onto ND via PEI 800 without background of other cellular populations. This antisense RNA specifically targets oncogenic miR-135b overexpressed in many tumors including mammary tumor cells such as 4T1. Once A135b reaches target miR-135b in cytoplasm, it drives its degradation. Hypothetically, decreasing the level of miR-135b impairs the growth of cancer cells and eventually leads to cell death. We have used cells treated with saline and cells treated with complex NDA135b (consisting of ND, transferrin conjugated with Alexa488, PEI800, and A135b). Using confocal microscopy, we monitored internalization of the NDA135b into the vast majority of adherent 4T1 cells (Figure 2a). The complexes that were just internalized or are about to be internalized are shown in yellow (merging color from red stained ND and green stained Alexa Fluor 488) and the image has been focused on the level of cellular nuclei (in blue) to distinguish internalized complexes.

After stimulation with NDA135b, the 4T1 cells were cultured for additional two days and at the end of the incubation period, we measured amount of lactate dehydrogenase, an enzyme release from damaged cells due to increased plasma membrane permeability. Performing this assay, we added several controls such as positive control (PC) representing cells with damaged plasma membrane; cells incubated with uncoated nanodiamonds (ND), cells incubated with ND coated with scrambled control RNA instead of specific A135b (NDSc), and cells incubated with a commercial transfection reagent loaded with A135b (A135CR). We found significantly increased (*p*-value < 0.01) amount of lactate dehydrogenase only in cells stimulated with A135b without difference in carrier (Figure 2b).

Increased permeability of plasma membrane points towards direct toxicity but it can also occur in the very late phase of programmed cell death such as apoptosis, especially in cancer cells monoculture lacking cells phagocyting apoptotic bodies. We have further tested percentage of cells undergoing early phases of apoptosis and we found that A135b, independently on carrier, significantly increased (*p*-value < 0.01) the percentage of apoptotic cells (Figure 2c). However, we have also found a slightly increased rate of apoptosis in cells incubated with ND and NDSc (*p*-value < 0.05), suggesting that exposure of these cells to ND affects their viability up to some extent.

We observed internalization of NDA135b into cells with confocal microscopy. To prove not only delivery of A135b but also its release into cytoplasm, we performed detection of target miR-135b localized in cancer cell cytoplasm by qPCR. We observed a significant decrease (*p* value < 0.01) in cells incubated with A135b delivered within NDA135b or with a commercial reagent (Figure 2d). There was no decrease in miR-135b level in cells incubated with ND or NDSc, proving the specific targeting of miR-135b with antisense A135b.

Data are presented as the mean ± standard deviation; * *p*-value < 0.05 and ** *p*-value < 0.01 versus saline-treated group (NC) were calculated by ANOVA.

### 3.3. Specific Internalization of NDA135b into Breast Cancer Cells Spheroids Co-Cultured with Peritoneal Lavage Cells

Adherent monocultures are great for obtaining basic characteristics of the material tested, however this setup lacks important features such as cell–cell communication and interaction and different special awareness found in in vivo conditions. To imitate the basic tumor microenvironment in vitro, we employed 3D mammospheres derived from 4T1 cells grown in conditions protecting them from the attachment to surface. The mammospheres were further co-cultured with cells obtained from peritoneal lavage of C57BL/6 mouse. The peritoneal lavage is rich in macrophages (in our samples, 45%–50% of leukocytes) that easily engulf nanoparticles and, thus, they are an important cell population to evaluate the cancer cell-targeting effectivity of transferrin adsorbed to NDA135b surface. In order to distinguish 4T1 cancer cells from peritoneal lavage cells (MF) when co-cultured, we loaded MF with cell tracking dye and it enabled us to visualize MF actively incorporating into 4T1 spheroids (Figure 3a).

To evaluate the effectivity of A135b delivered into our 3D tumor, we performed qPCR. We found that only samples incubated with A135b significantly decreased (*p*-value < 0.01) the level of cytoplasmic miR-135b (Figure 3b). In contrast to 4T1 cancer cells, the peritoneal lavage cells do not express miR-135b as shown in Figure A1.

4T1 cells or MF loaded with NDA135b can be also detected by flow cytometry due to transferrin conjugated with Texas Red. We used flow cytometry to quantify how many cancer cells and MF carried the NDA135b. The NDA135b was designed to carry transferrin as a targeting structure to facilitate the uptake of the particle by cancer cells expressing transferrin receptor and decrease the uptake by normal cells including phagocytes. We confirmed that in our system, most (98% ± 1.2) 4T1 cancer cells express transferrin receptor in contrast to 7% (6.6% ± 1.5) of cells present in peritoneal lavage (Figure 3c). When we checked the cells present in peritoneal lavage thoroughly, we found that 61% ± 0.5 of transferrin receptor positive cells belong to monocyte/macrophage fraction as shown in Figure 3d. Remaining populations expressed transferrin receptor from 5% to 12% of cells (8.9% ± 0.6 of erythrocytes, 12.1% ± 1.5 of lymphocytes, and 5.8% ± 0.5 of granulocytes).

In order to evaluate percentage of 4T1 cells and MF interacting with NDA135b within the tumor microenvironment, we incubated co-cultures with NDA135b and subsequently detected cells positive for transferrin. In parallel, we have incubated 4T1 spheroids and MF with ND135b separately to obtain controls of individual cell populations. While incubated alone, about 60% of 4T1 cells and about 37% of MF were positive for NDA135b (Figure 4a). The percentage of NDA135b-positive cells was calculated as positivity in both right quadrants. There is a distinct population within the MF sample (marked in right bottom quadrant) that represents about 6% of MF cells positive for NDA135b once the MF cells were incubated alone with NDA135b. This population represents live cells that carry NDA135b. We expected those cells to be macrophages present in MF population that engulfed NDA135b. Our presumption was based on the result mentioned in Figure 3d showing that in total, 4% of macrophages in our MF samples express transferrin receptor (calculated as 0.61*6.6%). To prove our hypothesis, we have gated the MF samples used for co-cultures for specific cell populations: Granulocytes, lymphocytes, and macrophages (Figure 4b). Subsequently, we tested each population for NDA135b positivity. Here, erythrocytes were not acquired due to optimal events distribution for further analyses of mixed 4T1 and MF cultures. We found that only about 30% (30% ± 1.5) of macrophages were positive for NDA135b. Surprisingly, we detected 65% of granulocytes (65% ± 3.2) and 14% (14% ± 1.7) of lymphocytes carrying NDA135b (Figure 4b).

### 3.4. Local and Systemic In Vivo Application of Targeted NDA135b

Systemic administration of nanoparticles represents major challenge for most researchers. Even constructs exhibiting great effectivity in vitro might encounter difficulties such as lower stability in blood environment, decreased circulation time in blood and fast renal clearance, or unspecific internalization and accumulation in off-target tissue. All of that contributes to low effectivity of the constructs. Regarding nanodiamond particles with the average core size of 70 nm used in this study, we are aware of problematic nanoparticles clearance once administered into the blood system without good targeting and shielding structures. In this study, we employed two different types of NDA135b administration-intratumoral (i.t.) application directly into tumor mass and intravenous (i.v.) application via tail vein. Our first goal was to compare the effectivity of both approaches in order to reach the tumor cells cytoplasm and eliminate target miR-135b. We found that after three consecutive doses (1 day apart), we could detect NDA135b in tumors excised from experimental animals (Figure 5a). Here, detection of NDA135b in tumor samples was enabled by using transferrin–Alexa Fluor 488 conjugate adsorbed on the nanodiamond core. We compared tumors from animals administered with NDA135b containing Alexa Fluor 488-labeled transferrin with animals that received saline (control animals) as well as animals that received NDA135b containing transferrin without Alexa Fluor 488 label.

To test the presence of nanoparticles inside the tumor cells, we performed flow cytometry analysis and qPCR-based detection of target miR-135b in tumor cells after tumor homogenization. We found significantly decreased levels of miR-135b in both groups—administered intratumoral or intravenous (Figure 5b). In parallel, we performed flow cytometry analysis to evaluate the percentage of tumor cells carrying NDA135b. We found 6.54% ± 0.3 cells positive for NDA135b in samples administered intratumoral (significant, ** *p*-value < 0.01) as shown in Figure 5c (bottom). This positivity was rather weak in animals with intravenous administration of NDA135b (1.43 ± 0.1; ns *p*-value = 0.34). Here, we also did not experience a well-defined positive peak, but rather consistent slight changes in dot plot distributions (Figure 5c; top).

### 3.5. Accumulation of Non-Internalized NDA135b Complex in Key Tissues after In Vivo Applications

Since we found touches of NDA135b in tumors even after intravenous administration, we wondered if NDA135b circulating in peripheral blood interacted with blood elements or accumulated in tissues such as kidney, liver, or spleen. We also wondered if NDA135b administered intratumorally could escape from the tumor site and accumulate in other tissue due to its size.

In our system, transferrin also represented a targeting molecule with high affinity to 4T1 cells expressing a high amount of surface transferrin receptor (Figure 3c) and, thus, it shielded the complexes from extensive non-specific uptake. We have analyzed excised organs from the control group as well as groups administered with NDA135b by a whole-body imaging system. However, we found no signs of NDA135b presence in any of the excised tissues (Figure 6a). We also performed flow cytometry analyses of homogenized tissues and we did not detect any positive signal pointing out the presence of NDA135b in examined samples (Figure 6b). Distinguishable sub-populations such as leukocytes are marked in the liver, kidney, and spleen dot plots (yellow). We checked for any positive signal within these subpopulations and they displayed similar signal intensities like the whole tissues (Figure A2). Within spleen samples, we also checked the erythrocyte/debris population (black gate) and there was no positive signal either (Figure A3). Interestingly we found very different splenocytes granularity using side scatter parameter (Figure 6c). The granularity of splenocytes was apparently lower in animals after intravenous administration of NDA135b in contrast to control animals and animals administered with NDA135b intratumorally (Figure 6c).

The intravenous administration of NDA135b exhibited an effect on splenocytes. It would be ideal if we could prove the final effect of A135b by qPCR. However, the splenocytes, as they are not cancerous cells, do not express miR-135b (Figure A1) so we cannot use this approach. Assuming an indirect effect on splenocytes and no accumulation in spleen, liver, or kidney, we examined the presence of NDA135b in bodily fluids such as blood and urine. We also checked basic parameters of major cell populations within peripheral blood. We tested the presence of NDA135b using fluorescence spectrometer and flow cytometry. In both fluids, urine and peripheral blood, we found significantly positive (*p*-value < 0.01) signals in samples from animals administered with NDA135b intravenously—compared to control animals (Figure 7a). We did not detect any positivity in fluids obtained from animals administered intratumoral. We analyzed basic blood cells populations—erythrocytes, monocytes/granulocytes, and lymphocytes. Nevertheless, we did not observe any positive signal in any of the mentioned subpopulations (Figure 7b). That suggests that NDA135b was present in peripheral blood but it was not internalized by any blood cells.

## 4. Discussion

In this report, we discuss, for the first time, the effectivity of HPHT nanodiamonds as carriers for short antisense RNAs when applied in 3D organoids as well as in vivo. Nanodiamonds prepared under HPHT possess great benefits like cyto-compatibility, cytoplasmic membrane penetration, easy surface decoration, or traceability, suggesting them for advanced biomedical use. Due to their size (in this report, an average of 70 nm), the HPHT nanodiamonds cannot be cleared from the system via renal filtration [28]. This may be a benefit since, if appropriately shielded, they can circulate in the blood stream for a longer time but on the other hand, they can accumulate in RES (reticuloendothelial system) organs. There is an abundance of particles without any specific structure that would facilitate their passage into target tissue for example tumor. Blood vessels of the majority of tumors exhibit enhanced permeability retention (EPR), supporting entrance of nanoparticles larger than 5 nm [29], so even particles without a targeting structure can reach the tumor tissue, but the efficacy is much lower [30].

Compatibility of nanomaterial with the bio-interface represents a key request every time the material is intended for any biomedical use. The compatibility evaluation primarily employs tests in vitro. So far, the only studies describing toxicity of nanodiamond particles employed detonation nanodiamonds with usual diameter 2–5 nm. Turcheniuk and Mochalin summarized that the possible cytotoxic effects reported in detonation nanodiamonds probably originated from usage of nanomaterial with a high degree of impurities [31]. Regarding HPHT nanodiamonds, all studies performed so far claimed no toxicity in vitro [32,33,34,35]. However, the surface modifications such as polyethylenimine (PEI) coating can affect the cyto-compatibility of the nanoparticle-polymer complex [36]. We support this finding as we have described it in our preceding study [13] too. The intermediate complexes consisting only of nanodiamond and PEI are not stable in colloids and tend to aggregate fast if a stabilizing component such as nucleic acid is not added immediately [13]. We suggested that large aggregates with positive total charge impair cellular membrane while adsorbing and entering the cells. In the current study, we have experienced slightly increased rate of early apoptosis (Figure 2c) in samples incubated with ND and NDSc, complexes without active A135b. The increase is rather negligible, yet statistically significant. We suspect the uptake of nanoparticles that reside in high concentration in 4T1 cell cytoplasm could interfere with the cell divisions. This hypothesis is supported by the results of direct toxicity (LDH release) shown in Figure 2b. Neither ND nor NDSc exhibited direct toxicity and significant release of LDH but the presence of those nanoparticles led, in a small percentage of cells, to apoptosis. When using NDA135b, the increase in apoptotic or dead cells was much more prominent (Figure 2b,c) due to A135b function (Figure 2d).

Unlike adherent 2D culture, the 3D tumor grown without surface attachment enables testing of nanocomplexes in an ambiance much closer to the natural situation in vivo. To imitate the tumor microenvironment and tests NDA135b specificity, we co-cultured the mammospheres with cells obtained from peritoneal cavity lavage of experimental animals. Nanodiamonds within NDA135b were decorated with transferrin in order to facilitate their uptake by tumor cells [37]. We believe it also promoted stabilization of the complexes and protected them from extensive binding of sera proteins. This hypothesis is based on recent studies that showed how adsorption of a specific protein onto ND before the exposure of ND to sera prevented formation of protein corona [38]. Decoration of ND surface with a protein such as Tf conjugated with a fluorophore increased the size of the final complex to about 300 nm and significantly increased the polydispersity index. The PdI increase might be due to uneven amount of conjugates linked to ND particle. This approach with fluorophore-labeled targeting structure is convenient for proof-of function presented in this study where we needed to control the complex location. If not necessary, we suggest using unlabeled targeting antibodies to obtain lower PdI.

The silencing of microRNA-135b in 4T1 cells by NDA135b performed in the 3D model was very effective (Figure 3b) suggesting good distribution of NDA135b in the media, which enabled its sufficient internalization into cells that are in a compact sphere not adhering and covering the bottom of the experimental dish. Flow cytometry analysis of 3D tumor spheres with intercalated cells from peritoneal lavage revealed very interesting information regarding cells carrying NDA135b. In this setup, we can benefit from existing communication between tumor and immune cells but the model is still much simpler than the situation in vivo. We expected the tumor cell will uptake most of NDA135b due to specific targeting of NDA135b with transferrin and shielding of the complex from macrophages. We believed macrophages from peritoneal lavage could up to some degree compete with tumor cells for the NDA135b since they are (in 7%) positive for transferrin receptor too. We found that the granulocytes are a major population that carries NDA135b (Figure 4). A presence of transferrin-receptor on the cell surface was ruled out by flow cytometry (Figure 3), thus, the granulocytes did carry NDA135b regardless the targeting molecule. So far, mostly monocyte/macrophage cells were studied regarding transportation of internalized nanoparticles within multicellular organism [39]. Now, we found granulocytes play an important role too and the future research regarding nanoparticle interaction should consider deeper study of this cell population. To our knowledge, there is only one report describing interaction of nanodiamonds with granulocytes specifically with neutrophils in air pouches of lungs [40].

Detailed reports describing the effect of HPHT nanodiamond particles on the immune system in vivo or complex organ toxicity are rare. So far, the non-targeted ND particles applied in vivo exhibited neither direct toxicity nor inflammatory and stress responses [23,24]. If nanodiamonds accumulated, predominantly in lungs, liver, or spleen, there were no symptoms of abnormalities [20,22]. Also, mice injected with ND did not show any weight-loss or other clinical signs of toxicity even after exposure for four weeks [22]. On the other hand, ND coated with hemagglutinin significantly enhanced the immunostimulatory effect of hemagglutinin in mice—however, it was more likely because of increased stability and concentration of the hemagglutinin than presence of nanodiamond carrier [41]. Modified ND in hydrosol applied intravenously into rabbits showed a short-term increase in serum bilirubin (sign of erythrocyte lysis) and other changes associated with the sequestering of the nanodiamonds in the liver [42].

As mentioned earlier specific coating and targeting of nanoparticles significantly effects their fate and final accumulation site [30]. Since we applied, in vivo, a newly developed construct (combination of RNA, protein, and polymer coating), we performed basic compatibility tests to assess particles’ fate and their interactions with biological structures. We wondered if localized application would trigger any side effects because of possible NDA135b leakage from tumor into peripheral blood system. Using intratumoral as well as intravenous application of NDA135b targeted with transferrin led to accumulation of NDA135b in the excised tumor mass (Figure 5a) and significant downregulation of cytoplasmic microRNA-135b. It shows that even the NDA135b circulating in blood after intravenous application effectively reached tumor cells. Once we isolated tumor cells in order to perform flow cytometry and quantitative estimate, we yielded a significantly different amount of NDA135b-positive cells concerning the type of administration. In samples after intratumoral application, we found a well-defined peak pointing out cells positive for NDA135b (Figure 5c). In samples after intravenous application, we did not see any specific peak but we still experienced some shift in cells’ positivity as shown in histograms (Figure 5c, top). We are aware the lower amount of cells positive for NDA135b in samples after intravenous application reflected the way of administration. However, the whole tumor (after intravenous application) exhibited quite strong fluorescence but the isolated tumor cells were insignificantly positive for NDA135b. We believe that after intratumoral application, NDA135b resided in the tumor site for a longer time, increasing its chance to be internalized. After intravenous administration, we assume that some circulating NDA135b reached the tumor and were internalized by tumor cells, but the amount of internalized complexes was too low to detect it. In addition, the non-internalized NDA135b within tumor mass were washed away during tumor cells isolation.

The lower amount of positive tumor cells after intravenous application suggested persisting of NDA135b in blood or its off-target accumulation. Our data show that NDA135b did not significantly accumulate in liver, kidney, or spleen, suggesting the coating prevented ND accumulation in those key tissues. In animals, after intravenous application of NDA135b, we found positivity in urine. Since the core particle is about 70 nm big, it is not possible that the fully coated complex would pass via renal filtration into urine [28]. There is a small chance that when the NDA135b entered blood cells, transferrin–fluorophore conjugate separated from the ND core and left the body via renal clearance. For example, dextrans conjugated with Texas Red are renal clearable [43].

Interestingly, analyzing splenocytes obtained from homogenized spleen samples, we found a remarkable shift in their granularity after intravenous application of NDA135b (Figure 6c). We suggested an indirect effect since we did not detect any presence of NDA135b in excised spleen tissues or different splenocytes populations. Assuming an indirect effect on splenocytes, we examined basic blood cells populations—erythrocytes, monocytes/granulocytes, and lymphocytes within peripheral blood. We expected to find positive signal in monocytes/granulocytes that could potentially interact with or internalize NDA135b. We found NDA135b-positive blood samples after intravenous application but we did not observe any positive signal in any of the cellular subpopulations (Figure 7). The data suggest that NDA135b was present in peripheral blood, it was not carried by any blood cells, but it interacted with a blood element to affect granularity of spleen tissue.

Within multicellular organism, an inflammatory reaction is a response to infection but also to non-infectious agents. Recently, Li et al. [44] described non-infectious inflammation after intraperitoneal application of hydrocarbon oil. This inflammation recruited neutrophils, dendritic cells, and macrophages into spleen based on elevated levels of interleukin-6 and tumor necrosis factor—alpha (TNF—α). Intravenous application of NDA135b could induce non-infections inflammation too. This hypothesis is supported by Munoz et al. [40]. The authors proposed that small (10 nm) naked or PEG-coated ND damage plasma membrane and trigger instability of lysosomal compartment and formation of neutrophil extracellular traps. It initialized inflammatory response comprising synthesis of pro-inflammatory cytokines. Larger particles (100 nm) seemed inert [40]. Our data (Figure A4) and other authors’ [22,23,24] reports reinforce the relative inertness regarding cytokine production too.

Comparing the two different in vivo administrations, the localized intratumoral application was effective in order to knockdown target microRNA-135b and exhibited no leakage of NDA135b into bodily fluids or other tissues. The intravenous application was effective in order to knockdown target microRNA-135b. It was accompanied with circulation of NDA135b in blood and it affected splenocytes parameters. According to our data, the intratumoral application represents a much safer mean of NDA135b application with much less side effects. We believe that localized and targeted application is the most effective and the safest approach regarding any biomedical use of non-degradable inorganic carriers with diameter exceeding limits for renal clearance. Keeping this approach in mind, we still can benefit from extraordinary characteristics of nanodiamond particles even though they are non-biodegradable. Thus nanodiamond-based complexes that are stabilized and decorated with nucleic acids and/or peptides are prospective in topical treatments of skin disorder such as chronic wound-healing, dermatitis, or skin tumors. Next to the therapeutic function (nucleic acid) performed only in particular cells (antibody), the nanodiamond core keeps its luminescence and with a suitable optical system, we could track the process of construct diffusion within the treated tissue. Using intravenous application of coated ND carriers requires further detailed analysis of cell interaction triggering changes in organ (spleen) characteristics. Since the immune response is dependent on carrier size as well as on way of administration [40], each ND carrier has to be evaluated with regard to its anticipated use. Importantly, our findings contribute to the understanding of ND carriers’ fate and trafficking in vivo. It has revealed new interesting interactions between ND carrier and biological interface as well as future challenges regarding signal transport and response mechanisms within in vivo systems.

## Figures and Tables

**Figure 1 nanomaterials-09-00866-f001:**
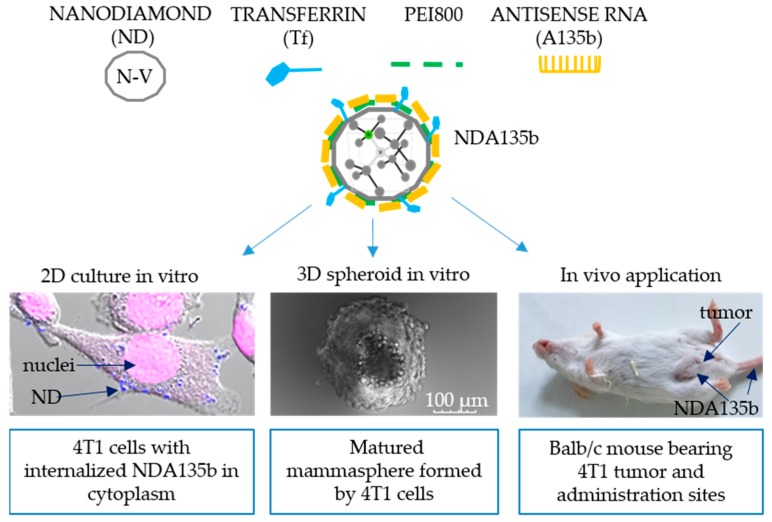
Composition of nanodiamond-based complex with multiple functionalization and its application onto a sequence of biological models from the easiest 2D cell culture via 3D differentiated mammospheres towards local and systemic in vivo administrations.

**Figure 2 nanomaterials-09-00866-f002:**
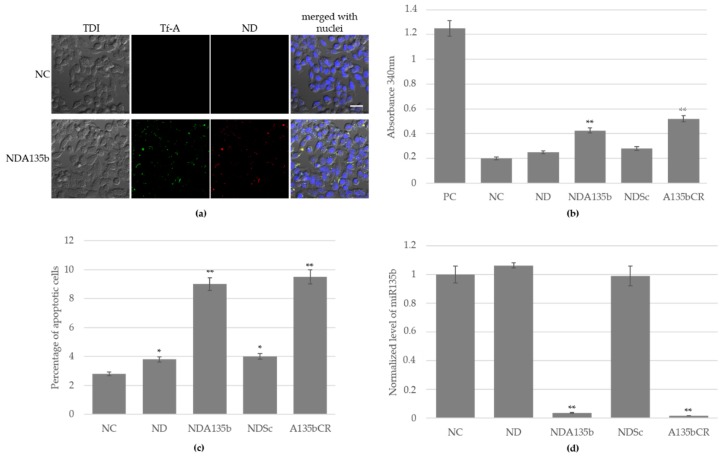
Internalization of functionalized nanodiamond carrier into 4T1 cells grown in 2D culture and the biological effect of antimiR-135b cargo: (**a**) Visualization of NDA135b, here decorated with transferrin conjugated to Alexa Fluor 488. Nuclei counterstained with Hoechst 33342 (blue). Fluorescence and transmission (TDI) visualized with a confocal microscopy Olympus FV1000, 40x. Scale bar is 20 µm; (**b**) release of lactate dehydrogenase indicating cell damage; (**c**) induction of apoptosis (early phase) measured with an AnnexinV assay; (**d**) detection of remaining miR-135b level in 4T1 cells after particular stimulation. Values of *p* ≤ 0.05 (*) and *p* ≤ 0.01 (**) were statistically significant.

**Figure 3 nanomaterials-09-00866-f003:**
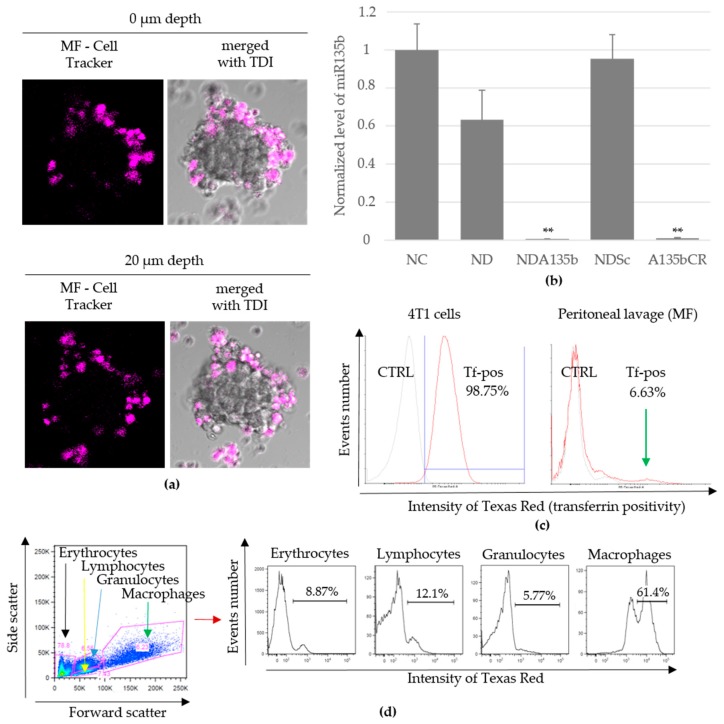
Targeting of 4T1 spheroids by functionalized nanodiamonds within a mixed 3D co-culture with peritoneal lavage cells imitating basic tumor microenvironment: (**a**) Determination of peritoneal lavage cells (MF, purple) integrating into matured mammosphere. Fluorescence and transmission (TDI) visualized with a confocal microscopy Olympus FV1000, 20×. Spheroid size is about 200 µm; (**b**) detection of remaining miR-135b level in mammospheres after particular stimulation. Data are presented as the mean ± standard deviation, ** *p*-value < 0.01 versus saline-treated group was obtained by ANOVA; (**c**) surface expression of transferrin receptor in 4T1 spheres (left) and peritoneal cells (right). CTRL stands for negative control of staining; (**d**) basic distribution of live cell sub-populations in peritoneal lavage and transferrin receptor positive cells detected within each sub-population.

**Figure 4 nanomaterials-09-00866-f004:**
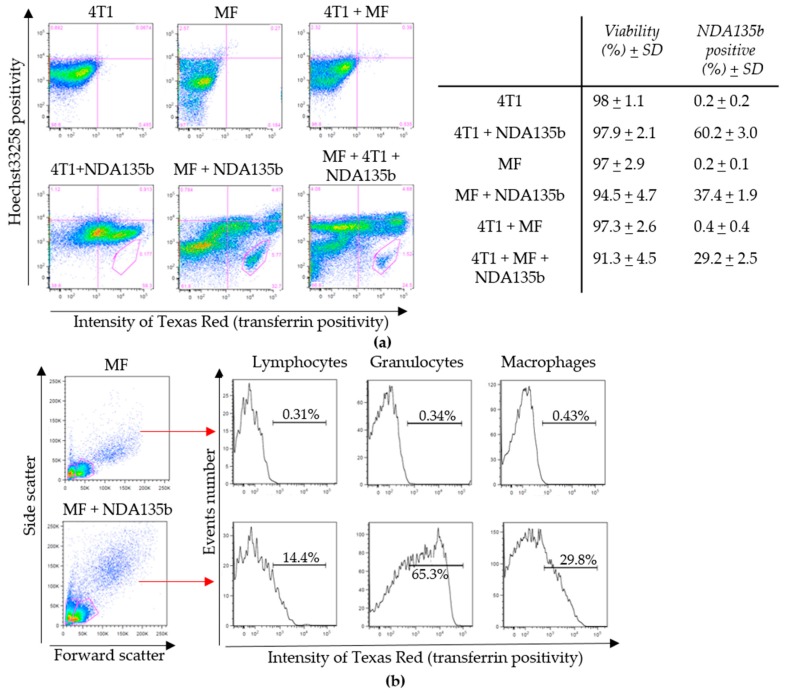
Targeting of 4T1 spheroids by functionalized nanodiamonds within a mixed 3D co-culture with peritoneal lavage cells (MF) imitating basic tumor microenvironment: (**a**) Uptake of targeted nanodiamond complex NDA135b by particular cells within a spheroid MF co-culture. The average percentage of positivity (and standard deviation) are listed on the right; (**b**) gating of live cells within the peritoneal lavage sample with or without stimulation by NDA135b (left) and detection of NDA135b in individual populations (right).

**Figure 5 nanomaterials-09-00866-f005:**
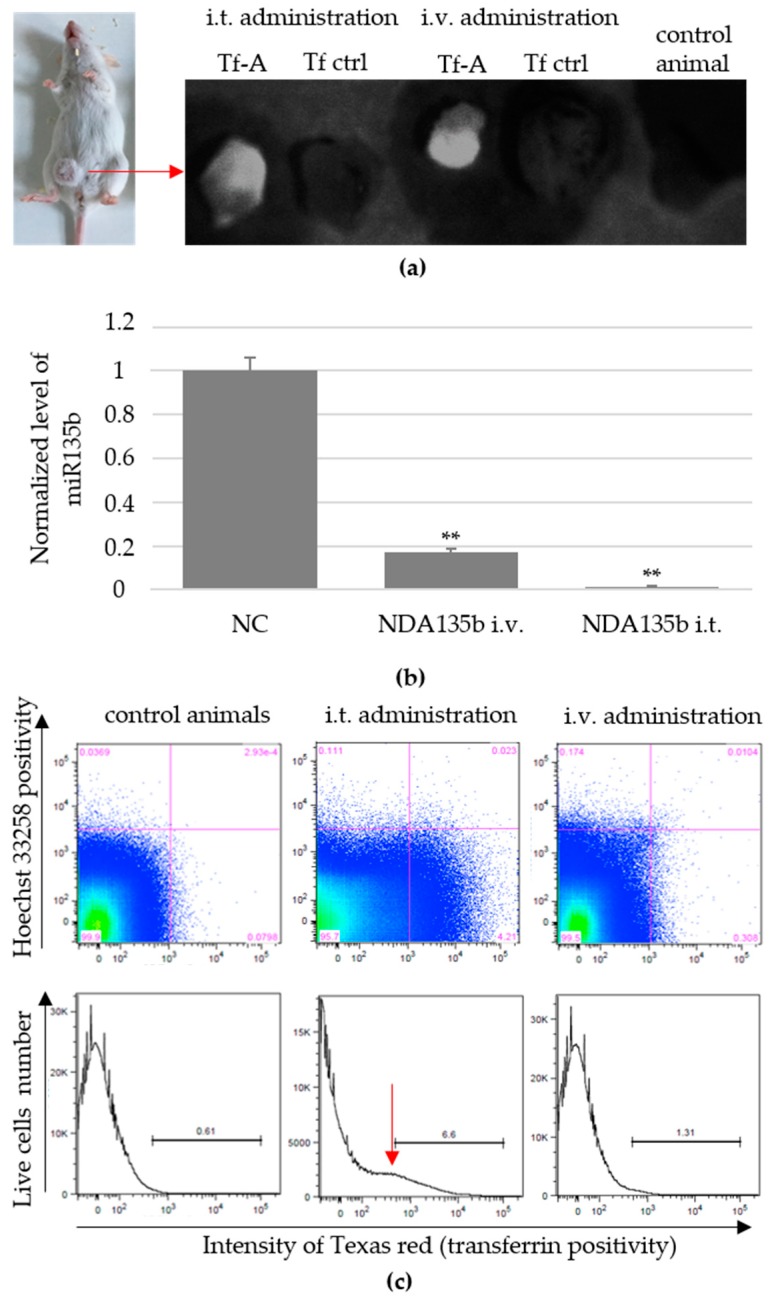
In vivo applications of NDA135b into breast tumor-bearing animals: (**a**) Detection of transferrin-Alexa Fluor 488 decorated NDA135b complexes (Tf-A) in tumor samples ex vivo. Tf ctrl stands for NDA135b decorated with Tf without the Alexa Fluor 488. Control animals received saline only; (**b**) detection of remaining miR-135b level in tumor after NDA135b administration. Data are presented as the mean ± standard deviation, ** *p*-value < 0.01 versus saline-treated group was obtained by ANOVA; (**c**) detection of NDA135b in cells obtained from excised tumors. Red arrow point to NDA135b-positive cells.

**Figure 6 nanomaterials-09-00866-f006:**
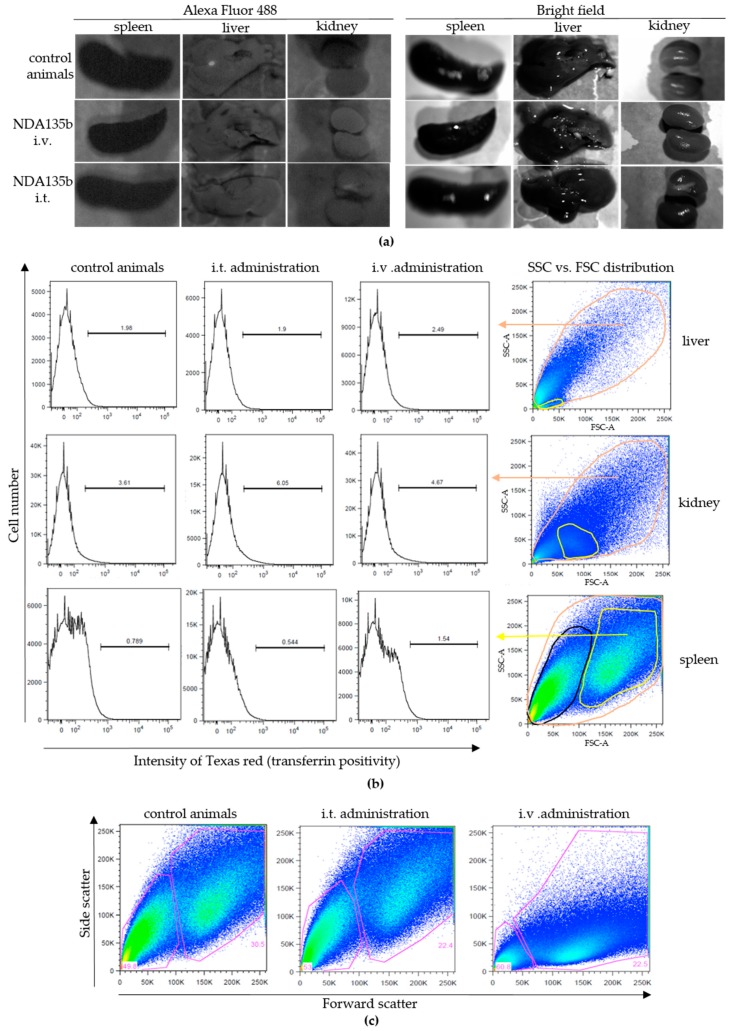
(**a**) Detection of transferrin conjugate in tissues excised from experimental animals administered with NDA135b intravenously or intratumoral: (**a**) Detection of NDA135b complexes in tissues ex vivo by microscopy. Control animals received saline only; (**b**) detection of NDA135b in excised tissues by flow cytometry. Representative dot plots and histograms are shown (exemplary gating is on the right). The histograms of liver and kidney samples consider signal from all cell-like events (orange gate). In spleen, histograms show signal in leukocytes (yellow gate in dot plots); (**c**) comparison of different splenocytes granularity after NDA135b intravenous and intratumoral applications.

**Figure 7 nanomaterials-09-00866-f007:**
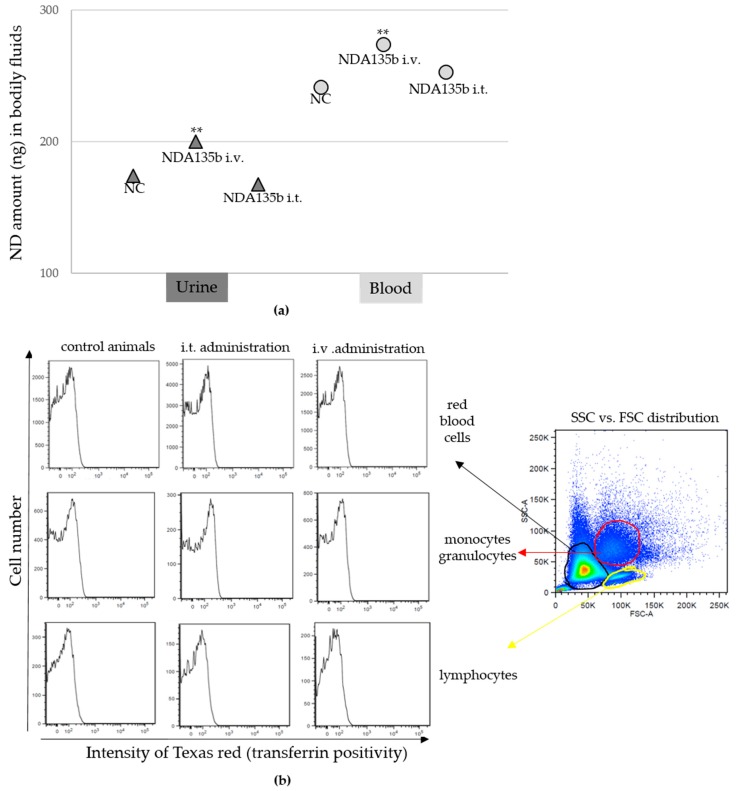
Detection of NDA135b decorated with transferrin-Texas red in bodily fluids obtained from tumor-bearing animals administered with ND-A135b intravenously or intratumoral: (**a**) Detection of transferrin–Texas red in acellular fraction of urine and blood. Data are presented as the mean ± standard deviation, ** *p*-value < 0.01 versus NC group was obtained by ANOVA; (**b**) detection of transferrin-Texas red in cellular fraction of peripheral blood. Representative dot plots and histograms are presented. The histograms show signal from different cell populations indicated in dot plot on the right side of the image (red blood cells, granulocytes/monocytes, and lymphocytes).

**Table 1 nanomaterials-09-00866-t001:** Basic characterization of NDA135b complex and its comparison to original nanodiamonds (ND) and similar complex ND-polyethylenimine (PEI)-RNA lacking targeting Tf-conjugate.

Sample	Zeta Potential (mV)	Average Size (nm)	PdI
NDA135b	−20 ± −3	353 ± 140	0.4
ND	−35 ± −7	73 ± 28	0.2
ND-PEI-RNA	−28 ± −4	120 ± 15	0.24

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
