# Peer review of "Knockdown of microRNA-135b in Mammary Carcinoma by Targeted Nanodiamonds: Potentials and Pitfalls of In Vivo Applications"

_nanomaterials, 2019, doi:10.3390/nano9060866_

Reviewer 1 Report

The required experiments/corrections have been duly performed. The manuscript can be accepted in this form.

Author Response

Comments and Suggestions for Authors: 

The required experiments/corrections have been duly performed. The manuscript can be accepted in this form.

Response: We thank you very much for you review. There are no comments that require to be answered. 

Reviewer 2 Report

Dear Editor,

In the resubmitted manuscript (Knockdown of microRNA-135b in mammary  carcinoma by targeted nanodiamonds: potentials &  pitfalls of in vivo applications) the authors answered point by point to all of the reviewer’s considerations.

It is evident that the authors spent time to answer in a proper way and to enhance the quality of their work. 

The added measurements are interesting.

Just one consideration:

Line 261: In the added table some typos. (please check the symbol ± in some errors I found just +)

The added considerations and perspectives are well written.

This paper is recommended for publication.

Author Response

Point 1:

Line 261: In the added table some typos. (please check the symbol ± in some errors I found just +)

Response 1: We thank you very much for the review. We have corrected symbols + for + in the added table on line 261.

This manuscript is a resubmission of an earlier submission. The following is a list of the peer review reports and author responses from that submission.

Round  1

Reviewer 1 Report

Dear Editor, the paper by Křivohlavá et al. entitled “Knock-down of microRNA-135b in mammary  carcinoma by targeted nanodiamonds: potentials & pitfalls of in vivo applications” comprehensively elucidates how localized intratumoral application of ND represents a suitable and safe approach for in vivo application of nanodiamond-based constructs. The subject fits to the journal scope and, in my opinion, is interesting for the scientific community. The paper is well written and the discussion at the end of the MS is clear. Brings the reader point by point to the conclusion, just a number of small Typos that are very simple to correct. From the scientific point of view I have no objections. Afterwards, there only several formal issues. Major requests: 1. Please add more perspectives opened by this study… would be very interesting to know how many problematics could be solved using this nanodiamonts. Minor requests: 2. Figures: Please improve the quality of Figure 3d and 4b…. not clear Based on the above arguments, I recommend the paper for publication after minor revision.

Reviewer 2 Report

The manuscript describes the development of transferrin-PEI-nanodiamonds containing miR135b and their in vivo efficacy. The rationale was provided but no details about the amount of bioactive compound retained by ND have been provided. Moreover, the amount of transferrin-derivatives integrated in the structure of nanosystems should be discussed because the concentration of protein can influence the binding features of a drug delivery system. Finally, the physico-chemical characterization of the modified ND should be investigated.

The manuscript cannot be accepted in this form.

Specific criticisms:

Page 2 – “Oxidized fluorescent nanodiamond ….and their preparation has been described in detail by Petrakova et al. and Lukowski et al. [13,26]”. The reference 13 is not useful because the composition of nanodiamonds is not discussed but only other references are reported. The materials used for the development of nanodiamonds should be briefly described in this section.

The integration of PEI and transferrin-derivatives within the ND structure is not clear. More details are required. It is not a chemical linkage and for this reason the amounts of the various components should be investigated by useful analytical methods.

Moreover, the amount of miR-A135b retained by the ND structure should be evaluated. In fact, in the biological experiments the concentration of the active compound (A135b) is required because it is the component able to exert the biological effect. Moreover, the empty formulations should be tested as reference when LDH assay is performed.

Page 4 – “Spectrometer measurements of NDA135b in fluids” – this section should be placed after “Animal model and in vivo NDA135b application”.

Page 4 - “Animal model and in vivo NDA135b application” – an empty formulation (ND without gene material) should be tested. A group treated with this formulation is fundamental in order to demonstrate the absence of pharmacological effects exerted by nanosystems.

Moreover, the sentence “The animals obtained 20μg of ND…” is not useful to provide information about the amount of A135b injected.

Page 5 – “3. Results” – a physico-chemical characterization of ND modified by using the various components, i.e. PEI, Transferrin, A135b, is fundamental. The mean sizes, polydispersity index and surface charge are required. The authors should compare the properties of ND described in the already published works with the those obtained in this manuscript. In particular, the use of PEI and gene materials could dramatically modulate the Zeta-potential of a formulation.

The results obtained by the LDH assay were not properly discussed.